# Strengthening laboratories in response to outbreaks in humanitarian emergencies and conflict settings: Results, challenges and lessons from expanding PCR diagnostic capacities for COVID-19 testing in Yemen

Ismail Mahat Bashir[1][⊙]*, Ali Ahmed Al-Waleedi[2][⊙]*, Saeed Mohamed Al-Shaibani[1][‡], Mohammed Rajamanar[1][‡], Shougi Al-Akbari[2][‡], Abdulelah Al-Harazi[3][‡], Layla Salim Aliwah[3][‡], Nahed Ahmed Salem[4][‡], Dina Al-Ademi[1][‡], Amal Barakat[5][‡], Nicole Sarkis[1][⊙], Abdinasir Abubakar[5][‡], Mikiko Senga[1][‡], Altaf Musani[1][‡], Adham Rashad Ismail Abdel Moneim[1][⊙], Nuha Mahmoud[1][⊙]*

1 World Health Organization, Sana'a, Yemen, 2 Ministry of Public Health and Population, Aden, Yemen, 3 National Centre for Public Health Laboratories, Sanaa, Yemen, 4 National Centre for Public Health Laboratories, Aden, Yemen, 5 World Health Organization, East Mediterranean Regional Office, Cairo, Egypt

⊙ These authors contributed equally to this work.
‡ These authors also contributed equally to this work.
* bashiri@who.int (IMB); ali_alwaleedi@hotmail.com (AAA-W); mahmoudn@who.int (NM)

## Abstract

### Background

When the COVID-19 pandemic was declared, Yemen, a country facing years of conflict had only one laboratory with PCR testing capacity. In this article, we describe the outcome of the implementation of molecular based diagnostics platform in Yemen and highlight the key milestones the country went through to increase access to testing for its populations residing in a geographically vast and politically divided country.

### Methods

A retrospective assessment of COVID-19 laboratory response activities was done detailing the needs assessment process, timelines, geographical coverage, and outcomes of the activities. Laboratory data was analyzed to construct the geographical locations of COVID-19 testing laboratories and the numbers of tests performed in each facility to highlight the demands of testing for travelers. Finally, we discuss the impact these activities had in enabling the movement of people across international borders for economic gains and in delivery of critical humanitarian aid.

### Outcome

PCR testing capacities in Yemen significantly improved, from one laboratory in Sanaa in April 2020 to 18 facilities across the country by June 2022. In addition, the number of functional Real-Time PCR thermocyclers increased from one to 32, the PCR tests output per

**Data Availability Statement:** All relevant data are within the manuscript and its Supporting information files.

**Funding:** The author(s) received no specific funding for this work.

**Competing interests:** The authors have declared that no competing interests exist.

day improved from 192 to 6144 tests per day. Results from analysis of laboratory data showed there were four peaks of COVID-19 in Yemen as October 2022. The majority of laboratory tests were performed for travelers than for medical or public health reasons. Demand for laboratory testing in Yemen was generally low and waned over time as the perceived risk of COVID-19 declined, in parallel with rollout of the COVID-19 vaccines.

## Discussion/Conclusion

The successful expansion of laboratory testing capacity was instrumental in the control and management of COVID-19 cases and critical in the implementation of public response strategies, including restrictions on gathering. Laboratory testing also facilitated the movement of humanitarian agencies and delivery of aid and enabled hundreds of thousands of Yemeni nationals to travel internationally. By virtue of these outcomes, the impact of laboratory strengthening activities was thus felt in the health sector and beyond.

## Introduction

In the first year of the emergence of SARS-COV2 and subsequent declaration as a public health emergency of international concern by WHO, control and prevention strategies were limited and mostly entailed prompt identification and isolation of infected persons to interrupt the transmission of the disease. Response pillars such as clinical management of cases with COVID-19 specific therapy, and other public health interventions such as vaccinations were unavailable. The importance of identifying infected persons placed a high demand on laboratory services–a nearly neglected pillar in the health system of many resource-limited countries–positioning these services at the forefront of the fight against COVID-19 [1].

Yemen, a country of 29 million people amidst armed conflict and outbreaks of other infectious diseases for several years, had one of the weakest laboratory systems in the WHO Eastern Mediterranean Region. In February 2020 –a month before the pandemic was declared, of Yemen's seven central public health laboratories (CPHL), only CPHL Sanaa had functional molecular PCR laboratory with the necessary capacity to conduct molecular assays using reverse transcriptase polymerase chain reaction (RT-PCR) technique. A laboratory is considered to have capacity to test for COVID-19 using PCR technology when it has at least three separate rooms for the different PCR functions, a functional PCR thermal cycler, running tap water, electricity, and a trained and competent laboratory workforce [2].

The declaration of the new outbreak was preceded by release of the genomic structure of SARS-COV2 which made it possible to identify genetic targets for developing PCR diagnostic tests [3]. As a novel entity in the catalogue of infectious diseases affecting humans, developing useful clinical tests was the immediate task of the scientific committee and bioscience companies who immediately embarked on the development of PCR diagnostic assays [4]. At this point, reverse transcriptase-polymerase chain reaction (RT-PCR) was the only diagnostic platform that could be used to identify cases. Owing to the increased demand for PCR diagnostic kits globally coupled with limited supply, there was an extreme global shortage for the few kits that were available then.

For Yemen, getting access to these diagnostic kits in the early stages of the pandemic was nearly impossible while capacity for designing in-house assays and production of primers were unavailable locally. As part of support to countries, a donation of two sets of desiccated

primers/probe mix targeting E-gene and RdRP specific genes for SARS-COV-2 was received from WHO Regional Office for the Eastern Mediterranean (EMRO). Owing to the political and administrative structure of Yemen, and considering the humanitarian principle of impartiality, one of the kits was donated to Sana'a and another for Aden–the seat of the internationally recognized government. While the laboratory in Sana'a having already a functional PCR unit performing molecular surveillance for influenza was able to immediately optimize the primer sets for COVID-19 PCR testing, the situation in Aden–with no PCR machine, no essential reagents like master mix, deoxyribonucleoside triphosphates (dNTPs), DNA polymerase, no plastics and other accessories, making use the desiccated primers and probes donated by WHO to provide testing for COVID-19 was nearly impossible. The situation was made worse by the fact that the two laboratories were situated in opposing ends of the conflicting parties in Yemen hence making it impossible to collaborate and utilize a sample referral system to cater for the needs of the people living in the ssouthern governorates of the country.

In this report, we highlight the ways in which the country recovered from an extremely limited capacity to provide RT-PCR testing for COVID-19 at the beginning of the pandemic to establish a network of molecular biology laboratories. We compare the core laboratory capacities for infectious disease diagnosis before and after the SARS-COV-2 pandemic and ways in which laboratory reports were used to guide the country's response to COVID-19. Finally, we illustrate how the need to facilitate the cross-border movement of Yemeni nationals surpassed the demand for testing for surveillance and case management purposes. Key challenges encountered during the implementation of laboratory activities laboratory and lessons that can be of important consideration if similar events occur in future are listed.

## Methodology

This study aimed to assess the laboratory capacity of Republic of Yemen before and after COVID-19 using a number of parameters including the number of facilities with technical capacity for PCR, PCR tests output per day, staff training/competency in PCR, utilization rates, and the number of conditions/diseases diagnosed by PCR among others. Important timelines and milestones in the project and in COVID-19 pandemic are illustrated to highlight ways in which laboratory readiness and strategic selection of the laboratories helped in identifying the index case of COVID-19 in Yemen. For a description of COVID-19 surveillance, data from the laboratories—accessed on 15 January 2023 was used to assess the temporal pattern of COVID-19 tests performed for surveillance purposes only in southern administrations of Yemen to identify the frequency of samples analyzed and the resultant peaks of cases that occurred in the country during the 3-year surveillance period. In addition, using data from all the PCR laboratories, we measured the frequency and location of tests from different laboratories to show the proportions of laboratory tests performed both for surveillance purposes and for travelers. We provide discussion on the capacity building and quality assurance processes that were implemented and their outcomes. Finally, we highlight the key challenges faced during the implementation of these activities and list a number of lessons that can be drawn from their implementation.

## Results

### Establishing and equipping the laboratory infrastructure

In a collaborative effort between the Ministry of Public Health and Population (MoPHP) and humanitarian partners, a prompt assessment was made to map the public laboratory resources present within the country and identify the infrastructural and instrumentation needs of each laboratory. Laboratory assessment tools developed by WHO were used to assess the status of

each of the laboratories [5]. Plans were designed to provide the necessary support to address infrastructural, technical, and operational gaps.

Based on the gaps identified in the assessment vis-a-vis the available resources—mainly donations from humanitarian partners, an initial plan for supporting the establishment of testing capacity in eight [8] laboratories with fully functional molecular biology section was proposed. In the weeks that followed, procurement of key laboratory equipment such as thermocyclers, other capital laboratory equipment such as freezers/refrigerators, consumables, and necessary personnel protective equipment (PPE) for the laboratories were initiated and completed. Establishing testing for the first laboratory in the seat of the Government was the most challenging. While the laboratory infrastructure was available in Aden CPHL, it did not have a functional thermocycler–the capital instrument necessary for conducting PCR test and the only one laboratory staff had attended previous training on PCR technique.

With collaborative efforts between MoPHP and international humanitarian partners, the first laboratory in the southern governorates was fully functional just 10 days after COVID-19 was declared a pandemic. This laboratory was set-up as an operational and capacity building centre for training of laboratory staff from the other planned laboratories. All training for laboratory staff was implemented by an in-country team of laboratory technical experts. At the end of the first 4 months, six of the eight laboratories planned were fully functional. The remaining two were delayed because of infrastructural limitations of the laboratory facilities that needed more time to construct. By early 2021, the laboratory strengthening investments that initially aimed for the diagnosis of SARS-COV-2 had successfully established 18 facilities with capacity for PCR. The geographical locations of these facilities (CPHL's and hospital laboratories) are shown in the map on Fig 1.

## Improvement in laboratory capacity in the country

Generally, there were significant improvements in laboratory capacity resulting from the implementation of laboratory strengthening activities between 2020 and 2022. PCR testing capacity that was initially confined to one laboratory in Sana'a became available in 18 facilities after one year of implementation. In addition, the number of functional RT-PCR thermocyclers increased from one to 32, the PCR tests output per day improved from 192 tests per day to 6144 tests per day, and the number of staff trained and competent in PCR work increased from four to 180. Most importantly, the number of diseases/conditions tested with RT-PCR technology increased as test kits for other diseases/conditions became available. Core laboratory capacities that improved through implementation of these activities are summarized in Table 1 below.

## Timelines for the establishment of testing laboratories vis-à-vis COVID-19 events

Implementation of health projects in complex emergencies is always challenging and this laboratory strengthening project was no different. In addition to all the operational challenges, "time" was also of the essence as COVID-19 was spreading quickly in the region. By the time the first recorded index case appeared in Yemen on 10th April 2020 in the coastal city of Mukalla in Hadramaut governorate, the country had three functional laboratories as indicated in the timeline in Fig 2. Being a point of entry by air, a seaport where international shipping lines dock, and populace regarded to have acumen for international trade, Mukalla was prioritized for developing PCR capacity at the initial stage of the implementation and this plan proved prudent.

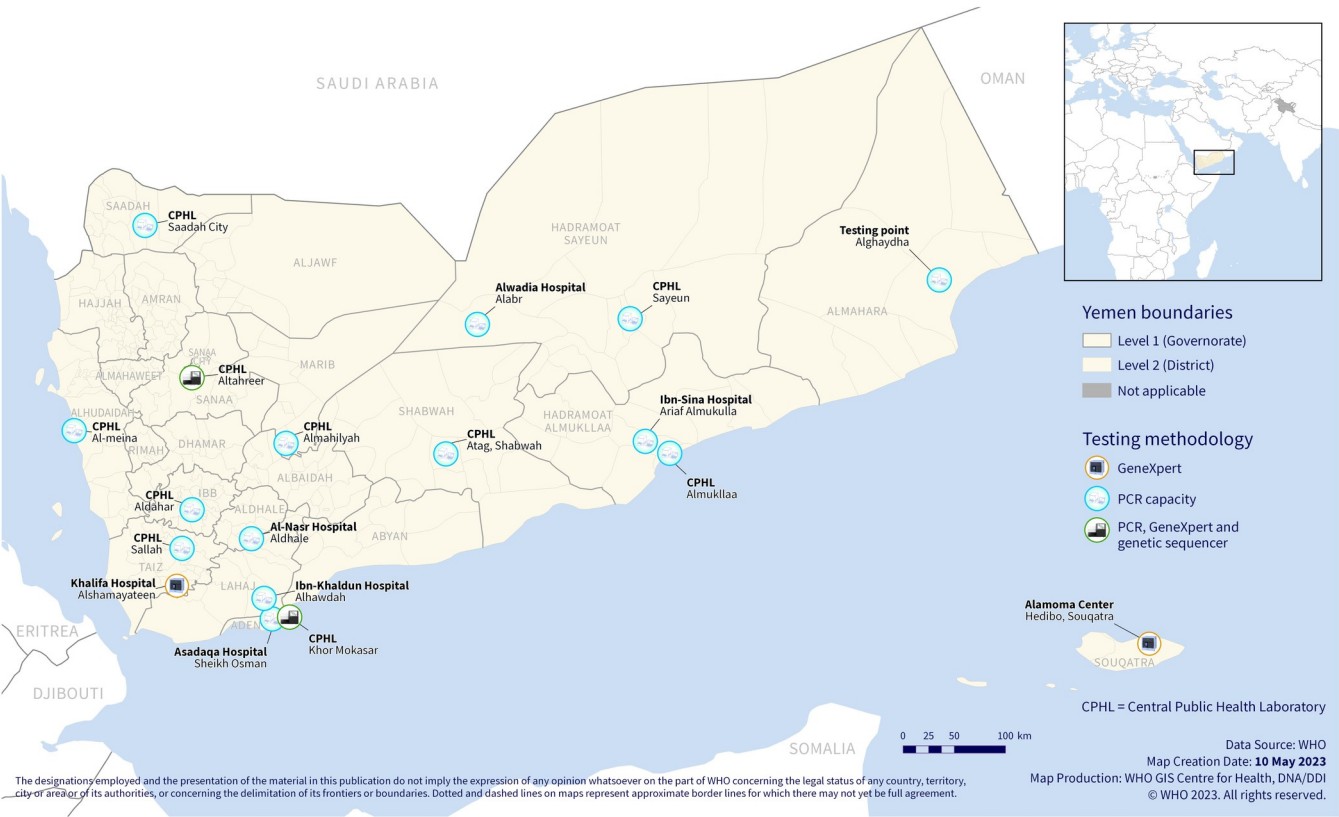

**Fig 1. Geographical locations of 18 laboratories with PCR, GeneXpert, and genetic sequencing capacity in Yemen by November 2022.**

## Numbers and temporal evolution of COVID-19 tests performed overtime for surveillance purposes

Using weekly data from the laboratory, the temporal patterns of the numbers of tests conducted over a three-year period was assessed. In total, 750,296 laboratory diagnostic tests for

**Table 1. Improvement in the molecular biology capacity in Yemen—Before and after the COVID-19 pandemic.**

| Intervention | Beginning of COVID-19 pandemic | December 2022 |
|---|---|---|
| Number of laboratories with functional PCR testing capacity | 1 | 18 |
| Number of functional thermocyclers | 1 | 32 |
| Testing capacity using PCR | 192/day | 6144 tests/day |
| Number of gene expert machines | 9 | 14 |
| Number of laboratory staff trained and competent in PCR technique | 4 | 180 |
| Diseases and conditions tested with PCR techniques | 1 – Influenzae surveillance | 8 – Influenzae, SARS-COV2, Dengue, Diphtheria, WNV, Chikungunya, Malaria, Hepatitis B. |
| Total number of COVID-19 tests done | | 750296 |
| • For surveillance | | • 44,406 (5.8%) |
| • Travelers | | • 706461 (94.2%) |
| Genomic sequencing capacity | 0 | 2 |

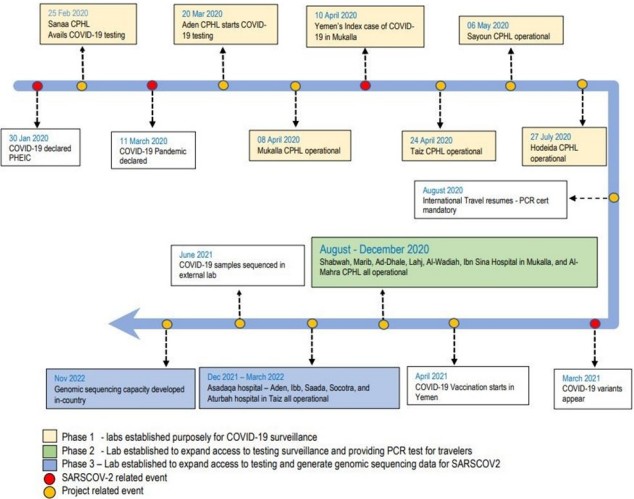

**Fig 2. Timelines highlighting important milestones on the evolution of virus in Yemen and period for the establishment of PCR and other laboratories.**

COVID-19 were performed from mid-April 2020 to December 2022. The breakdown of the total tests performed indicates only 44,406 tests were performed for surveillance purposes which accounted for 5.8% of all tests done for COVID-19 in the country. Out of the total number done for surveillance, 11,313 tests (25%) were conducted in 2020, while 25,753 (58%) occurred in 2021, dropping to 7434 (16%) in 2022. The temporal pattern of COVID-19 testing shows the emergence of four distinct waves of COVID-19 in the country as shown in Fig 3.

## The main reasons for COVID-19 testing

We also aimed to know the main reasons PCR testing was performed in the country and proportions of tests done for surveillance purposes vis-a-vis for travelers. On average, the number and proportions of tests done for travelers was 706,461 (94%) of all tests done. Demand was highest in the testing facilities located at the borders with Saudi Arabia and in cities where international flights resumed. Two laboratories–Aden CPHL located in Aden city (with an international airport) and Al-Wadia Hospital in Al-Abr district of Hadramaut (a land crossing

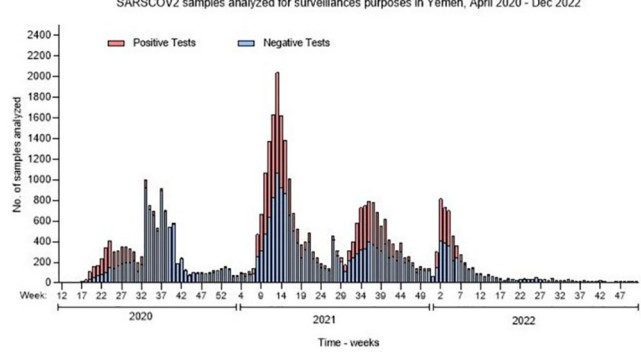

**Fig 3. Temporal patterns of SARS-COV-2 samples analyzed for the purposes of surveillance in Yemen from April 2020 to December 2022.**

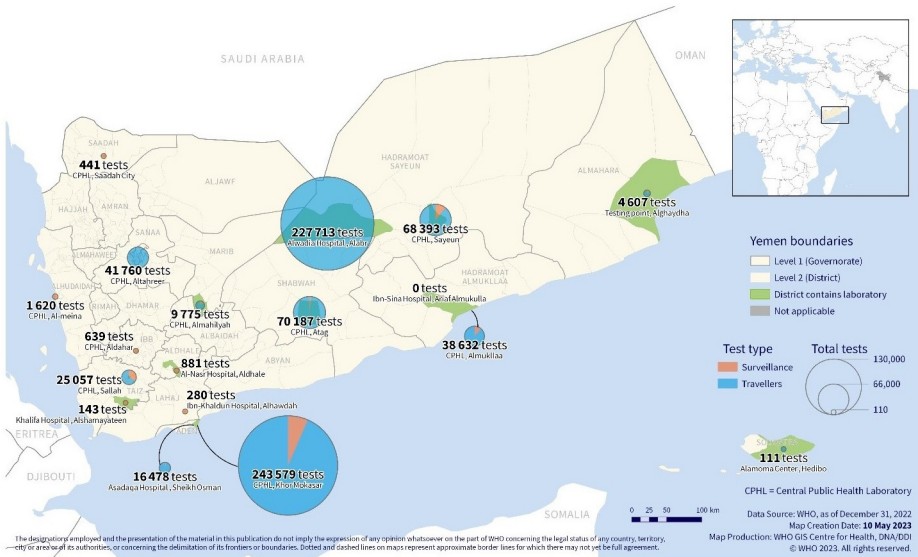

**Fig 4. Location and volumes of PCR testing performed for surveillance purpose (orange colour) and for travelers (blue) in Yemen by December 2022.** More testing happened in the southern governorates of the country where points of entry (POEs) are located, noting no positive COVID-19 cases were formally reported from the northern governorates.

point to Saudi Arabia)–performed the bulk of the testing. In the northern governorates, a large proportion of tests were done in Sana'a CPHL situated in Sana'a city and where many humanitarian agencies are operationally based. Since the de-facto authorities in Sana'a did not report COVID-19 cases, the tests done in Sana'a were mainly for humanitarian travelers. The number and proportions of PCR tests done in all laboratories are illustrated in Fig 4.

## Capacity building for laboratory staff

Unlike other laboratory techniques, RT-PCR tests are a technically demanding and complex laboratory procedure that entails different steps of RNA isolation, master-mix preparation, PCR processing in a thermocycler, and interpretation of the report. Extensive staff training was required to develop a competent laboratory workforce in all the stages of testing. For Yemen, to ensure staff competency in this new technique, laboratory technologists with at least an undergraduate degree were selected to undergo training and perform tests afterward in the laboratories. A team of experts in molecular biology supported by WHO and other programs, adopted the training module developed by FIND to support countries that was used for training laboratory staff. At least five different training sessions were conducted for laboratory staff. Trainings took a total of 5-days each and covered topics including biosafety, infection control and prevention, nasopharyngeal sample collection, PCR process, and data collection and sharing. In total, 180 laboratory staff went through the training and were deployed to the testing laboratories between 2020 and 2022. In addition, WHO invested in a two-week training on Genome Sequencing and Bioinformatics Analysis delivered by the International Livestock Research Institute (ILRI) in Nairobi, Kenya for seven Yemeni laboratory experts (71% female). Further capacity-building on genetic sequencing is being organized on priority basis to make the most of the investments in this area.

## Quality assurance procedures

The reliance on laboratory results depends importantly on the quality assurance procedures in place. RT-PCR testing for COVID-19 was to some extent deficient in many ways due to the limited knowledge people had on clinical and immunological evolution of the virus in the human body [6]. There was a need to reduce the chances of error in the laboratory as much as possible so that the deficiency in the sampling or the lack of clear knowledge of the virus clinical evolution was not compounded by errors in the laboratory technical procedures or in the interpretation of amplification curves, or in the determination of cut-offs for the cycle thresholds. To ensure compliance to the quality demands of the testing, SOPs were developed for all procedures and all staff performing tests were trained on them. In addition, laboratory staff were trained to focus on the individual assay controls to determine the acceptability of each test procedure performed. As a rule, the performance of at least three internal controls were required to declare an assay as successfully passed. Technical support was also provided to the laboratories by a senior molecular biologist who performed supportive supervision and on-the-job training. Finally, WHO supported the participation of two main laboratories in an annual external quality assurance (EQA) scheme and six laboratories in a sub-national EQA system. The performance of these laboratories over a three-year period is summarized in the Table 2 below.

## Discussion

The existence of one laboratory with PCR capacity in the country at the beginning of the pandemic was signaled as a weak link in the COVID-19 response plan given the geographical vastness of the country [7,8]. There was an urgent need to support the MoPHP to expand the laboratory capacities and extend testing to geographically distant governorates including the archipelago island of Socotra or areas cut-off by the raging conflict. The only option for the health actors was to support the establishment of testing capacity for COVID-19 using RT-PCR–the recommended diagnostic platform for the confirmation of cases at that time [9] The initial plan envisioned in MOHS Yemen COVID-19 response plan was to extend testing capacity to eight laboratories. However, as more donors got involved, the plans were continuously reviewed so that capacity was expanded to reach as many governorates of Yemen as possible, eventually leading to establishment of a network of the 18 testing laboratories. The scaling-up of testing from one to 18 sites in a country with innumerable challenges in terms of security, a fragile healthcare system, difficult logistics, and general complexity in implementation of health programs was a significant achievement. The use of both regular PCR devices and GeneXpert machines enabled the expansion of testing facilities. GeneXpert machines were deployed in governorates where the laboratory facilities lacked the space required for a PCR laboratory, or in laboratories that lacked essential equipment such as biosafety cabinets or had insufficiently qualified laboratory staff. For these facilities, GeneXpert was essential as it is an

**Table 2. Performance of two national-level laboratories in external quality assessment scheme.**

| Performance of two laboratories in EQA | | |
|---|---|---|
| Period | Number and percentage of SARS-COV-2 samples correctly identified | |
| Laboratory | Sana'a CPHL | Aden CPHL |
| 2020 | 5/5 (100%) | 5/5 (100%) |
| 2021 | 5/5 (100%) | 5/5 (100%) |
| 2022 | 5/5 (100%) | 5/5 (100%) |

EQA was organized by WHO and Public Health Laboratory Services–Hong Kong SAR (China)

easier to use technology and requires less physical space (only a single laboratory bench). Because it is a form of nucleic acid-based testing, it technically provides the same diagnostic outcome as a PCR test in terms of sensitivity and specificity. GeneXpert also does not expose laboratory staff to risks as it is a closed system. However, owing to exorbitant cost of the COVID-19 test cartridges and the global limited stocks available GeneXpert was deployed only selectively based on need.

Despite the high numbers of COVID-19 tests performed in Yemen, the numbers of tests performed for the purposes of surveillance and case management made up a minority of the total tests. The number of laboratory tests performed for surveillance was generally low throughout the pandemic period. This trend in low testing numbers and low utilization rates for COVID-19 tests was also seen in countries with similar settings to Yemen such as Libya, Somalia and Syria [10–12]. The low testing numbers recorded were not comparable to the population and the expected testing needs in the country. In the early phase of the pandemic, as COVID-19 gained a foothold in Yemen, morbidity was increasing rapidly, and case fatality rates were the highest reported globally standing at 17% compared to a global average of 2% [13–15]. This was in part due to the low denominator of tests conducted, with cases primarily being detected on presentation to facilities when the disease progression was already moderate to severe.

Furthermore, many hospitals and healthcare workers especially in private facilities were turning away patients ill with other conditions for fear of contracting COVID-19 while others were subjecting patients to radiological examinations (in the absence of PCR testing in many private facilities) as a means of ruling out COVID-19 and keeping their facilities and staff safe. Despite this reality, the uptake of testing and utilization of the PCR laboratories did not match the expected testing needs in the country for the management and control of COVID-19. Testing hesitancy among health workers and the general population, initial restrictions of COVID-19 tests that prioritized the elderly, people with comorbidities, and healthcare workers, and non-prioritization of COVID-19 in the northern governates are some of the reasons that resulted in the low utilization rates for COVID-19 tests in Yemen.

Once PCR laboratories were operational, data from laboratories were submitted to MoPHP central surveillance units seamlessly to enable early detection of peaks and tracking of the evolution of the epidemic. In many countries, tracking the evolution was essential in many ways as it determined other public health measures such as determination of closure of schools, reduction of meetings/social gatherings among other public health measures [16]. As a result, reports from laboratories were essential for making decisions on when to trigger public health preventive actions. There were four significant COVID-19 peaks in the country in the three-year period. The number of peaks in cases was lower than the average of number of peaks reported by countries in the Eastern Mediterranean Region which stood at five and at the global level shows at least six peaks [17]. While the low testing rates may have contributed to the fewer peaks recorded in Yemen compared to the region, we speculate that the travel embargo embossed on Yemen nationals during the conflict and low inflow of non-Yemeni nationals to the country may have reduced the chances of introduction of new strains of COVID-19. Furthermore, humanitarians were the only non-Yemeni nationals going to and from the country. Humanitarians working in Yemen strictly practiced pre- and post-travel PCR testing for COVID-19, subjected a mandatory quarantine on arrival back to Yemen for all staff under the strict supervision of a UN clinic, and were all vaccinated once the COVID-19 vaccines were available.

As COVID-19 restrictions eased and international travel partially resumed the demand for PCR testing significantly increased, based on the need to display a negative test result to gain entry or exit international borders. For Yemen, this demand for testing by travelers dwarfed

the needs of the health sector for the purposes of surveillance and clinical management. It is estimated that around 2 million Yemeni nationals work in Saudi Arabia in formal and informal jobs. Most of these nationals returned to Yemen when the pandemic was declared and international lockdowns imposed. Once international travel resumed, there was a mass exodus of Yemeni nationals back to Saudi Arabia to resume their gainful employment and this exodus created a huge demand for COVID-19 testing services. Due to the contributions of these nationals working in Saudi Arabia to the economy of the fragile country, the government through MoPHP felt obligated to facilitate the establishment of testing along the international borders and POEs. Hence, focus shifted away from COVID-19 laboratory testing and surveillance and priority was given to serving the needs of travelers. MoPHP strategically established some laboratories for this purpose and engaged bilaterally with some donors for laboratory capacity improvement. Capacity building of laboratory staff, implementation of quality assurance checks, and issuing of digitized PCR test certificates to enhance credibility of the tests ensured unhindered passage of Yemeni citizens back to Saudi Arabia.

Unlike other interventions such as treatment or vaccinations, the health impact of diagnostics on population health is usually difficult to assess [18]. In the present example, the outcome of the strengthened diagnostics went beyond the health sector as it was critical to the opening of the economy and the delivery of vital aid through facilitating the movement of humanitarians. Molecular based testing commonly referred to by many as polymerase chain reaction (PCR) was unavailable in many resource-limited countries before COVID-19 despite its invention in 1985 by Karry Mullis and its potential as diagnostic platform [19]. The need for existing laboratory infrastructure such as 3 separate rooms for the different PCR functions, a functional PCR machine (thermal cycler), running tap water, reliable electricity, and trained and competent laboratory workforce were some of the reasons why the test was not commonly available in resource-limited settings before the SARS-COV-2 pandemic. We argue that unlike other laboratory techniques such as conventional bacteriology methods, enzyme-linked immunesorbent assays, or other conventional molecular techniques for the diagnosis of infectious diseases, RT-PCR based assays provide diagnostic accuracy that is unmatched by other platforms and can actually be made available in resource-limited settings to improve disease diagnostics and enhance preparedness and response to outbreaks. With PCR diagnostic kits for almost all diseases of importance currently available commercially, initial set-up is the only hindrance for adoption of the technology. Once set-up is complete and the laboratory workforce is trained, the platform can significantly improve laboratory readiness and provide accurate diagnostic testing for both infectious and non-infectious diseases.

Using the heightened global interest in COVID-19, Yemen was able to utilize the opportunity to invest in material infrastructure and expand access to testing by molecular based platforms for the diagnosis of both COVID-19 and non-COVID-19 infectious diseases. This investment will remain useful for the country in the years to come and will be an important foundation upon which to improve disease detection and surveillance.

This report has a number of limitations in regards to the COVID-19 surveillance. The COVID-19 epidemiological trends presented including the peaks of the outbreak depicts only the status of outbreak in South governorates under the jurisdiction of the Internationally Recognized Government and not Yemen as a whole. The de facto authorities ruling the capital–Sanaa and the Northern governorates where majority of the Yemen's population reside did not consider COVID-19 as a public health threat and hence health facilities were not allowed to test symptomatic patients or perform any form of surveillance for SARS-COV2. In addition, the laboratory strengthening programs including the one reported in this article that aimed to establish new laboratory facilities to support COVID-19 surveillance and response were not accorded the necessary goodwill and administrative support. This lack of political buy-in

hindered implementation and delayed the establishment of COVID-19 testing in laboratories located in the Northern governorates. Finally, the numbers and trends of COVID-19 testing among travelers presented in this report are biased in the sense that they only present the tests done in public health laboratories. We are aware of the existence of private laboratories who were providing testing for travelers. Owing to the subsidized costs offered in government-supported public health laboratories and the reduced turn-around times for getting results back, most travelers opted to do their testing in public health laboratories rather than private facilities.

## Key challenges

Some of the main challenges encountered during the implementation of the project and in providing laboratory support are listed in Table 3.

**Table 3. Challenges encountered during the pandemic response in Yemen Challenges.**

| Challenge | Description |
|---|---|
| Logistics for samples and consumables | Logistical challenges related to shipment of clinical samples from one governorate to another existed both within and outside Yemen. Movement of essential items between the northern and southern governorates is frequently affected by long delays due to chain of clearances of cargo required by different authorities. Occasionally, the program had to contend with a complete cessation of movement of goods for several months enforced by different authorities especially in contested jurisdictions. Moreover, bringing supplies to Yemen from outside was even more difficult due to limited cargo airlifts. This situation led to delay in delivery of some essential supplies and temporary cessation of testing in some laboratories. For diagnostics and reagents, the delays in the logistic chain meant that by the time some PCR diagnostic kits arrived to laboratories in Yemen, they had lost 50% of the usable shelf-life–usually of 12 months for PCR test kits. |
| Low motivation of laboratory staff | Healthcare workers in Yemen are some of least motivated owing to many reasons such as non-payment or delayed payment of salaries, working in resource-limited environment, and working for a longer duration of time, among others. These factors were exacerbated by the COVID-19 pandemic and the inability to cope with an initial spike in cases and deaths. Furthermore, unlike doctors and nurses who received periodic incentives from humanitarian agencies, laboratory staff were not included as beneficiaries of such incentives. These factors led to low motivation among laboratory staff affecting provision of 24-hour services especially at the beginning of the pandemic. |
| Low demand for testing services | The demand for COVID-19 testing in Yemen for clinical and surveillance purposes was one of the lowest in the Eastern Mediterranean Region. In the early stages, due to the lack of specific treatment for COVID-19, people with clinical signs similar to COVID-19 preferred to ignore their status to avoid the stigma associated with the disease and to avoid compulsory isolation. Hence, the number of tests performed for surveillance and clinical purposes remained low even after access to laboratory testing improved significantly.<br>To increase demand for testing, Yemen also explored expansion into community-based testing using rapid diagnostics, however access challenges owing to the conflict, combined with limited availability of rapid diagnostic supplies and lower sensitivity when tested in the field, and misinformation and stigma around COVID-19, hindered these efforts. |

*(Continued)*

**Table 3.** (Continued)

| Challenge | Description |
|---|---|
| Prioritization of COVID-19 as a health problem in some parts of Yemen | In the northern governorates of Yemen where about 70% of the population lived, the de-facto authorities considered COVID-19 to be of no public health significance. Hence, response activities such as improving surveillance, developing testing laboratory capacity, or enhancing preparedness at the health facilities were relegated to the bottom of their health priorities. Moreover, programs that aimed to support COVID-19 response we not accorded support. While implementing laboratory strengthening activities, the lack of political buy-in hindered implementation and delayed the establishment of testing in laboratories such as Ibb and Sa'ada. |
| Genomic sequencing capacity | Lack of capacity for genomic sequencing within Yemen was a key challenge affecting the identification of circulating variants of SARS-COV-2: To our knowledge, by the end of 2022, Yemen had sequenced only 28 cases of SARS-COV-2 virus–one of the lowest globally. The lack of genomic sequencing capacity and the logistical nightmares associated with international shipment of infectious agents were mainly the reason for the lower genomic testing in the country. However, as sequencing and application of genomic data in disease epidemiology are expected to play crucial roles in disease detection and surveillance, having local capacity will be of paramount importance [20]. Laboratories in Aden and Sana'a have now introduced sequencing capacity. If coordinated properly, such efforts will ensure that wet laboratory procedures can be carried out in Yemen to generate genomic data. Such data can then be shared with collaborative agencies such as international research centers and academia in different parts of the world for bioinformatics and pipeline analysis to generate useful public health information until the country develops a competent and scientific-oriented workforce. Such collaboration should focus on knowledge transfer to enhance the technical capacity of local scientists in areas of genomics, bioinformatics and analytical pipelines for selected pathogens such SARS-COV-2, cholera and others relevant to the context of Yemen [21]. |

## Lessons learned

| Subject | Lessons learned |
|---|---|
| Logistics of perishable reagents | In contexts where there are limited shipping options, laboratory programs must balance the lead time that could be more than six months in some instances and the quantities of items to procure. Because of the challenges of procurement and logistics, there is a tendency to procure bulk quantities. In such cases, the long duration of lead time and the short shelf-life of PCR diagnostic kits and other reagents can lead to wastage of resources. |
| Shipment of clinical samples | Sample shipment to an external laboratory from areas in conflict usually takes more than four months from the time of initial arrangement to the actual shipping outside the country. This duration is unsuitable in emergency situations. In addition, unavailability of dried ice from the locality also affects the sample integrity. The lengthy turn-around time to obtain results means that they can only provide retrospective outcomes but will be of limited clinical or epidemiological significance. As a result, in-country capacity is key in detecting and responding timely and appropriately to the threats of infectious diseases. |

(*Continued*)

| Reliance on external support for testing and capacity improvement | The first lesson that was learnt from COVID-19 and from the implementation of these activities is the fact that when there is a universal threat to the health and well-being of populations affecting all countries globally, international solidarity can no longer be relied upon and everyone for themselves comes into play. In such situations, the primary goal of countries will be to meet their fundamental obligation of protecting their populations first and preventing the collapse of their healthcare systems. Hence support to neighboring countries irrespective of their urgent needs will naturally be relegated to a tertiary objective. In such instances, having self-sufficiency and developing national capacities to detect diseases is key and will lead to better outcome in the control of outbreaks. |
|---|---|
| Multisectoral engagement | In Yemen, there was generally a multisectoral approach to challenges such as movement of key equipment and supplies. Support from humanitarian agencies such as World Food Programme was essential in prioritizing the movement of essential diagnostics and laboratory consumables in areas where the Programme frequently operated. |

## Conclusion

In resource-limited settings with a high burden of infectious diseases, the need for timely response to any disease's threats are key to saving lives and reducing disease burden. Experience from Yemen has proven that it is possible to deliver complex technical support such as set-up of molecular diagnostics in a relatively short period of time. The return on investment from such ventures has already been shown by the role it played at the height of the COVID-19 pandemic and in enabling cross-border movement, delivery of aid, and enhancing disease detection. However, more returns will be seen on the use of this platform for detecting all forms of infectious diseases in future if some funding to maintain its functionality is implemented. It is imperative for governments through MoPHP, development partners, and humanitarian actors to prioritize support to sustain the full functionality of this investment to enhance detection and response to infectious disease outbreaks at the local level which will in turn bolster health security at the global level.

## Supporting information

**S1 Dataset.**
(XLSX)

## Acknowledgments

We would like to acknowledge the support of WHO GIS Centre for Health for support in the production of the maps in this article.

Laboratory strengthening activities in Yemen reported on in this study, were carried out with financial support from the World Bank to WHO via the Yemen COVID-19 Response Project (YCRP) (2020–2022) and the Emergency Health and Nutrition Project (EHNP) (2017–2022). Funding from the Islamic Development Bank (IsDB) and King Salman Humanitarian Aid and Relief Center (KS Relief) also contributed to the activities described in this study.

## Author Contributions

**Conceptualization:** Ismail Mahat Bashir, Ali Ahmed Al-Waleedi, Nicole Sarkis, Abdinasir Abubakar, Mikiko Senga, Altaf Musani, Adham Rashad Ismail Abdel Moneim, Nuha Mahmoud.

**Data curation:** Ismail Mahat Bashir, Mohammed Rajamanar, Layla Salim Aliwah, Nahed Ahmed Salem, Nicole Sarkis.

**Formal analysis:** Ismail Mahat Bashir, Ali Ahmed Al-Waleedi.

**Methodology:** Ismail Mahat Bashir.

**Supervision:** Altaf Musani, Adham Rashad Ismail Abdel Moneim, Nuha Mahmoud.

**Writing – original draft:** Ismail Mahat Bashir.

**Writing – review & editing:** Ismail Mahat Bashir, Ali Ahmed Al-Waleedi, Saeed Mohamed Al-Shaibani, Mohammed Rajamanar, Shougi Al-Akbari, Abdulelah Al-Harazi, Layla Salim Aliwah, Nahed Ahmed Salem, Dina Al-Ademi, Amal Barakat, Nicole Sarkis, Abdinasir Abubakar, Mikiko Senga, Altaf Musani, Adham Rashad Ismail Abdel Moneim, Nuha Mahmoud.

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
