## [Decision Letter · Decision Letter 0]

17 Oct 2023

PONE-D-23-28143Strengthening laboratories in response to outbreaks in humanitarian emergencies and conflict settings: results, challenges and lessons from expanding PCR diagnostic capacities for COVID-19 testing in Yemen.PLOS ONE

Dear Dr. BASHIR,

Thank you for submitting your manuscript to PLOS ONE. After careful consideration, we feel that it has merit but does not fully meet PLOS ONE’s publication criteria as it currently stands. Therefore, we invite you to submit a revised version of the manuscript that addresses the points raised during the review process.

We look forward to receiving your revised manuscript.

Kind regards,

Mathumalar Loganathan Fahrni

Academic Editor

PLOS ONE

Journal Requirements:

3. We note that Figures 1 and 4 in your submission contain [map/satellite] images which may be copyrighted. All PLOS content is published under the Creative Commons Attribution License (CC BY 4.0), which means that the manuscript, images, and Supporting Information files will be freely available online, and any third party is permitted to access, download, copy, distribute, and use these materials in any way, even commercially, with proper attribution. For these reasons, we cannot publish previously copyrighted maps or satellite images created using proprietary data, such as Google software (Google Maps, Street View, and Earth). For more information, see our copyright guidelines: http://journals.plos.org/plosone/s/licenses-and-copyright.

a. You may seek permission from the original copyright holder of Figures 1 and 4 to publish the content specifically under the CC BY 4.0 license.  

Additional Editor Comments:

Dear Authors,

The manuscript highlights the increasing need for molecular-based diagnostics and the processes which Yemen had underwent to achieve greater capacity for laboratory testing.

Could the authors kindly resubmit after writing the assessment report using the format of a strengths–weaknesses–opportunities–threats (SWOT) analytical method.

Regards,

Plos One Editorial Team

Reviewers' comments:

Reviewer's Responses to Questions

**Comments to the Author**

1. Is the manuscript technically sound, and do the data support the conclusions?

Reviewer #1: Yes

Reviewer #2: Yes

2. Has the statistical analysis been performed appropriately and rigorously? 

Reviewer #1: Yes

Reviewer #2: Yes

3. Have the authors made all data underlying the findings in their manuscript fully available?

Reviewer #1: Yes

Reviewer #2: Yes

4. Is the manuscript presented in an intelligible fashion and written in standard English?

Reviewer #1: Yes

Reviewer #2: Yes

5. Review Comments to the Author

Reviewer #1: This study aimed to evaluate the laboratory capacity of the Republic of Yemen before and after the Corona virus

117 Using a number of parameters including the number of facilities with technical capacity for PCR, PCR

Output of 118 tests daily, staff training/proficiency in PCR, utilization rates, no

119 cases/diseases diagnosed by PCR and others. Timetables and important milestones in

120 projects and in the COVID-19 pandemic highlighted the ways in which laboratory preparedness and

Strategic selection of 121 laboratories helped identify the index case of COVID-19 in Yemen. The article is very clear and has importance for Yemen and even for the world in clarifying the technical and economic problems facing these countries.

Reviewer #2: The topic included Yemen as a whole, while the laboratories in which Covid tests are available are in some governorates, so why was it not specified in the topic, for example, the main governorates in Yemen?

6. PLOS authors have the option to publish the peer review history of their article (what does this mean?). If published, this will include your full peer review and any attached files.

Reviewer #1: **Yes: **Hassan Abdulwahab Al-Shamahy

Reviewer #2: No

---

## [Author Response · Author response to Decision Letter 0]

11 Nov 2023

Journal Requirements:

Revised according to the PLOS ONE's style requirements

Minimal underlying data set submitted as supporting information and is attached 

3. We note that Figures 1 and 4 in your submission contain [map/satellite] images which may be copyrighted. All PLOS content is published under the Creative Commons Attribution License (CC BY 4.0), which means that the manuscript, images, and Supporting Information files will be freely available online, and any third party is permitted to access, download, copy, distribute, and use these materials in any way, even commercially, with proper attribution. For these reasons, we cannot publish previously copyrighted maps or satellite images created using proprietary data, such as Google software (Google Maps, Street View, and Earth). For more information, see our copyright guidelines: http://journals.plos.org/plosone/s/licenses-and-copyright.

a. You may seek permission from the original copyright holder of Figures 1 and 4 to publish the content specifically under the CC BY 4.0 license. 

A permission from the original copyright holder of Figures 1 and 4 – attached as other documents

Additional Editor Comments:

Dear Authors,

The manuscript highlights the increasing need for molecular-based diagnostics and the processes which Yemen had underwent to achieve greater capacity for laboratory testing.

Could the authors kindly resubmit after writing the assessment report using the format of a strengths–weaknesses–opportunities–threats (SWOT) analytical method.

Regards,

Plos One Editorial Team

We acknowledge the current format of the report may seem suitable to present it using SWOT methodology. We believe that we will have another opportunity to perform the SWOT assessment of the laboratory systems in Yemen. However, for the current report we prefer to retain in its current format as an article. 

Reviewers' comments:

Reviewer's Responses to Questions

Comments to the Author

1. Is the manuscript technically sound, and do the data support the conclusions?

Reviewer #1: Yes

Reviewer #2: Yes

2. Has the statistical analysis been performed appropriately and rigorously? 

Reviewer #1: Yes

Reviewer #2: Yes

3. Have the authors made all data underlying the findings in their manuscript fully available?

Reviewer #1: Yes

Reviewer #2: Yes

4. Is the manuscript presented in an intelligible fashion and written in standard English?

Reviewer #1: Yes

Reviewer #2: Yes

5. Review Comments to the Author

Reviewer #1: This study aimed to evaluate the laboratory capacity of the Republic of Yemen before and after the Corona virus

117 Using a number of parameters including the number of facilities with technical capacity for PCR, PCR

Output of 118 tests daily, staff training/proficiency in PCR, utilization rates, no

119 cases/diseases diagnosed by PCR and others. Timetables and important milestones in

120 projects and in the COVID-19 pandemic highlighted the ways in which laboratory preparedness and

Strategic selection of 121 laboratories helped identify the index case of COVID-19 in Yemen. The article is very clear and has importance for Yemen and even for the world in clarifying the technical and economic problems facing these countries.

No comments were required 

Reviewer #2: The topic included Yemen as a whole, while the laboratories in which Covid tests are available are in some governorates, so why was it not specified in the topic, for example, the main governorates in Yemen? 

Comments to reviewer #2: The project intended to support laboratory strengthening in the Yemen as a whole. The names and locations where the services are available are highlighted in the Fig 1. Majority of the governorates either had a PCR laboratory facility and we preferred the main topic to highlight that - covering all Yemen. However, we have included statements on the limitation of the report in regards to the this and specifically highlighting that the data presented were from specific governorates that were submitting data in regards to COVID-19 testing. 

6. PLOS authors have the option to publish the peer review history of their article (what does this mean?). If published, this will include your full peer review and any attached files.

Do you want your identity to be public for this peer review? For information about this choice, including consent withdrawal, please see our Privacy Policy.

Reviewer #1: Yes: Hassan Abdulwahab Al-Shamahy

Reviewer #2: No

---

## [Decision Letter · Decision Letter 1]

29 Jan 2024

Strengthening laboratories in response to outbreaks in humanitarian emergencies and conflict settings: results, challenges and lessons from expanding PCR diagnostic capacities for COVID-19 testing in Yemen.

PONE-D-23-28143R1

Dear Dr. Ismail Mahat Bashir,

We’re pleased to inform you that your manuscript has been judged scientifically suitable for publication and will be formally accepted for publication once it meets all outstanding technical requirements.

Kind regards,

Sana Eybpoosh

Academic Editor

PLOS ONE

Additional Editor Comments (optional):

Reviewers' comments:

Reviewer's Responses to Questions

**Comments to the Author**

1. If the authors have adequately addressed your comments raised in a previous round of review and you feel that this manuscript is now acceptable for publication, you may indicate that here to bypass the “Comments to the Author” section, enter your conflict of interest statement in the “Confidential to Editor” section, and submit your "Accept" recommendation.

Reviewer #2: All comments have been addressed

2. Is the manuscript technically sound, and do the data support the conclusions?

Reviewer #2: Yes

3. Has the statistical analysis been performed appropriately and rigorously? 

Reviewer #2: Yes

4. Have the authors made all data underlying the findings in their manuscript fully available?

Reviewer #2: Yes

5. Is the manuscript presented in an intelligible fashion and written in standard English?

Reviewer #2: Yes

6. Review Comments to the Author

Reviewer #2: This article has been revised and ready to be published and used in future research, and I believe it will be shared in many research centers.

7. PLOS authors have the option to publish the peer review history of their article (what does this mean?). If published, this will include your full peer review and any attached files.

Reviewer #2: **Yes: **Sami Mohammed Abdo Hassan

---

## [Editor Report · Acceptance letter]

14 Feb 2024

PONE-D-23-28143R1 

PLOS ONE

Dear Dr. BASHIR, 

I'm pleased to inform you that your manuscript has been deemed suitable for publication in PLOS ONE. Congratulations! Your manuscript is now being handed over to our production team.

Kind regards, 

on behalf of

Dr. Sana Eybpoosh 

Academic Editor

PLOS ONE